# Macrolide-Resistant *Mycoplasma pneumoniae* Among Japanese Children from 2008 to 2024

**DOI:** 10.3390/microorganisms13102243

**Published:** 2025-09-25

**Authors:** Tomohiro Oishi, Tsuyoshi Kenri, Daisuke Yoshioka

**Affiliations:** 1Department of Clinical Infectious Diseases, Kawasaki Medical School, 577 Matsushima, Kurashiki 701-0192, Japan; 2Department of Bacteriology II, National Institute of Infectious Diseases, Japan Institute for Health Security, Tokyo 208-0011, Japan; kenri@niid.go.jp

**Keywords:** *Mycoplasma pneumoniae*, macrolide-resistance, children

## Abstract

*Mycoplasma pneumoniae* (MP), an important pathogen that causes pneumonia among children and young adults, caused an epidemic every four years in Japan until 2016, with the next epidemic occurring eight years later in 2024. This study compared the prevalence of MP infections among Japanese children in 2024 to previous years using real-time polymerase chain reaction (PCR) and the *p1* genotype determined using the PCR restriction fragment length polymorphism typing method from nasopharyngeal swab specimens. Of the 133 total isolates collected in 2024, 54.1% were macrolide-resistant MP (MRMP), with 98.0% of those containing an A2063G mutation in the 23S rRNA gene associated with macrolide resistance. This annual rate of MRMP and incidence of the A2063G mutation was similar to those in 2016. However, the dominant *p1* genotype among isolates in 2024 was type 1 (93.4%), whereas type 2 was dominant in the previous epidemic. Thus, although the rate of MRMP in 2024 was similar to that in the previous epidemic year, the distribution of *p1* genotypes was different. Further, the rate of MRMP was lower than neighboring Asian countries, including China and Korea, but was higher than in European countries. Therefore, it is important to continue monitoring MP infections in Japan.

## 1. Introduction

*Mycoplasma pneumoniae* (MP), a bacterium that lacks a cell wall [1], is a major pathogen that causes community-acquired pneumonia, especially among children and young adults, with the proportion of cases caused by MP increasing with age [2,3,4,5,6]. Endemic MP infections reportedly occur repeatedly every 4–7 years [7]. In Japan, an epidemic of MP infections occurred every four years from 1960 to 1988, coinciding with the Olympic games, and was termed “Olympic Pneumonia” or “Olympic diseases” [8]. This trend disappeared after the introduction of new macrolide antibiotics, such as clarithromycin and azithromycin. However, macrolide-resistant MP (MRMP) began to emerge in early 2000 [8,9] and an MP epidemic occurred in 2011. Further, the proportion of MRMP among all MP cases increased, reaching 81.8% in 2012 [10]. Since then, there had been an MP epidemic every four years, and the rate of MRMP was closely correlated with the number of MP infections in Japan. Furthermore, P1 genotyping of MP isolates, which encodes the P1 protein of *M. pneumoniae*, also periodically changed from the type 1 to the type 2 lineage in Japan [11]. However, there was a pause in epidemics caused by MP infections starting after the epidemic in 2016, which included the years of the coronavirus disease (COVID-19) pandemic. This remained the case until an MP epidemic occurred in Japan in 2024, marking the first epidemic in eight years. Given this long interval since the previous epidemic, we investigated the current epidemic of MP infections and MRMP in Japan.

## 2. Materials and Methods

### 2.1. Ethical Aspects

The study protocol was approved by the Ethics Committee of Kawasaki Medical School, Kurashiki, Japan, on 14 December 2024 (no. 3119-06).

### 2.2. Materials

We have collected MP samples since 2008 from the nasopharynx of children from 0 to 15 years of age suspected of having a lower respiratory tract infection due to MP. The cooperating medical settings consisted of 85 facilities, 50 clinics, and 35 hospitals, which were located in the Kyushu, Chugoku-Shikoku, Kinki, Kanto-Chubu, and Tohoku-Hokkaido regions in Japan. In 2024, the samples were collected from eight medical settings, two clinics, and six hospitals, in Kyushu, Chugoku-Shikoku, Kinki, and Kanto-Chubu areas in Japan. The enrolled children included both outpatients and hospitalized patients under 15 years of age with suspected lower respiratory infections due to *M. pneumoniae*. Pediatricians at the respective facilities collected nasopharyngeal swabs or sputum samples from children with respiratory tract infections. The nasopharyngeal swab specimens were transported at room temperature within two days to our laboratory at Kawasaki Medical School via a courier service.

### 2.3. Real-Time Polymerase Chain Reaction Analysis

Sterile swabs (JCB Industry Limited, Tokyo, Japan) used to collect respiratory samples from children were placed into 3 mL of universal viral transport medium (Becton, Dickinson and Company, Sparks, MD, USA) and transported at 20 °C within two days to Kawasaki Medical School. For the detection of MP, real-time polymerase chain reaction (PCR) was performed on DNA extracted from nasopharyngeal swab specimens, as previously mentioned [10]. Specifically, crude DNA was extracted using the following procedure: 300 μL of resuspended transport medium was transferred into a 1.5 mL microtube and centrifuged at 4 °C at 20,000× *g* for 30 min. Thereafter, 285 μL of the supernatant was discarded and the remainder was transferred into a thin-walled 200 μL PCR tube that contained 85 μL of lysis buffer [2 mmol/L Tris-HCl (pH 8.3), 10 mmol/L KCl, 0.045 mmol/L MgCl_2_, 0.45% Triton X-100, 0.45% Tween 20, and 0.4 mg/mL RNA-grade Proteinase K (Thermo Fisher Scientific Inc., Waltham, MA, USA)]. This suspension was mixed via gently pipetting, incubated at 55 °C for 60 min, incubated at 100 °C for 10 min, and then cooled to 4 °C. MP DNA was detected via real-time PCR targeting a conserved region of the gene encoding the P1 adhesin [1]. A direct sequencing method was used to determine the nucleotides at positions 2063, 2064, and 2617 in domain V of the 23S rRNA gene, which are associated with resistance to macrolides.

### 2.4. P1 Genotyping of MP Isolates

*P1* genotyping of MP isolates was determined using a combination of PCR restriction fragment length polymorphism (PCR-RFLP) typing and nucleotide sequence analysis of the polymorphisms in *p1*, as previously described [12]. Specifically, genomic DNA was extracted from the MP isolates collected in culture medium using the QIAamp DNA Mini Kit (Qiagen, Tokyo, Japan). Two primer pairs (ADH1-ADH2 and ADH3-ADH4) were used to amplify the polymorphic regions in the *p1* gene (containing RepMP4 or RepMP2/3 regions) via PCR amplification of the genomic DNA. PCR products were digested with restriction enzymes *Hae*III or *Mbo*I, and separated via electrophoresis on a 2% agarose gel. After staining the gel with ethidium bromide, the RFLP patterns of the PCR products were compared to determine the *p1* type.

## 3. Results

### 3.1. Patient Characteristics

In total, 188 samples were collected. Among them, 163 *M. pneumoniae* strains (83.3%) were determined to be positive via PCR and 133 (70.8%) were cultured. Among the cultured isolates, some were from children (77 boys and 56 girls) seen at eight medical settings from April to December 2024. The median age and interquartile range were 8 years and 5–9 years, respectively (Figure 1).

### 3.2. Distribution of MRMP Due to a Mutation in the 23S rRNA Gene Isolated from Japanese Children of Various Age Groups in 2024

Table 1 shows the distribution of MRMP and 23S rRNA gene mutations among various age groups of Japanese children in 2024. The overall prevalence rate of MRMP was 54.1%, and all MRMP isolates had an A2063G mutation, except one MRMP isolate with a C2617A mutation. Although macrolide-resistant rates were higher among younger age groups, there was no significant difference among groups.

### 3.3. Number of MP Infections and the Detection Rate of Mrmp Infections in Japan by Year

The overall prevalence rate of MRMP in Japan from 2008 to 2024 is shown in Figure 2. From a total of 5443 samples, 2111 (38.8%) were detected to be positive via PCR, and 1221 (22.4%) could be cultured. There was an increased number of MP isolates in 2011, 2012, 2015, 2016, and 2024. The MRMP rate increased in 2011 and peaked at 81.8% in 2012. Thereafter, the rate of resistance followed a similar trend to the number of MP isolates. The rate in 2024 (54.1%) was similar to the rate observed in 2016 (65.3%), which was the most recent endemic prior to 2024.

### 3.4. Detection Rate of Mrmp Infections and the Distribution of Point Mutations in All MRMP Isolates by Year

As shown in Table 2, 96.5% of all MRMP isolates throughout the years had an A2063G mutation in the 23S rRNA gene that is associated with macrolide resistance. Additional mutations in the 23S rRNA gene, such as A2063C, A2063T, A2064G, C2617G, and C2617T, were found at much lower rates.

### 3.5. Annual Distribution of MP Isolates Based on p1 Genotyping

Figure 3 shows the frequency of p1 genotyping for the MP isolates from 2008 to 2024. Genotyping was determined using PCR-RFLP analysis, as previously described [8]. In 2024, the prevalence of the p1 gene type 1 among 16 randomly selected strains was 93.4% (15/16). The distribution of the p1 genotype has changed from year to year. Specifically, the p1 genotype 1 was predominant during the endemic in 2011 and 2012, but the p1 genotype 2 subtypes were predominant during the next endemic in 2015 and 2016.

### 3.6. Annual Macrolide-Resistance Rate Among p1 Genotype Type 1 and 2 Isolates

Annual macrolide-resistance rates among MP isolates with p1 genotypes 1 and 2 are shown in Figure 4. In 2024, 66.7% of MP isolates determined to be genotype 1 were macrolide-resistant, with a resistance range from 53.4% to 100% from 2008 to 2024. However, the resistance rate for MP isolates determined to be p1 genotypetwo subtypes ranged from 0% to 25% over the same time period.

## 4. Discussion

In our study, the incidence of MP infections in 2024 was greatest in the 5- to 9-year-old group and the macrolide-resistance rate decreased with increasing age. This trend is similar to those previously reported that showed that MP was more common in children ≥5 years [6,9]. Miyashita et al. reported that the macrolide-resistance rate among MP isolates from adult patients in our hospital from 2008 to 2011 was 41.1% (30/73) [13]. This rate is lower than that observed among MP isolates from children, which was 74.8% (853/1140) during that same time frame in this study. In another report, Xie et al. reported a frequency of MRMP of 41.7% (30/72) in adult patients from 2011 to 2017 in Beijing, China [14], whereas the macrolide-resistance rate was reportedly 66.4% (142/214) among MP isolated from children in Beijing, China, by Dou et al. [15]. Thus, the macrolide-resistance rate of MP tends to be lower in adults than in children. Although not significant, the rate of macrolide resistance among MP isolates from younger children was higher in our study. Also, although a difference in the rate of macrolide resistance due to age group is unclear, age may be a factor of MRMP production.

Next, the detection rate of MRMP in 2024 was similar to the rate in the previous epidemic in 2015–2016, but higher than those after the last epidemic. Our data since 2008 demonstrates that the rate of MRMP is related to the number of MP isolates. This trend was the same for the data from 2024. Further, the rates of MRMP in recent years were higher in east Asia than those reported in this study. Specifically, in 2023, Chen et al. reported a 100% rate of MRMP in Beijing, China [16], and Chin and Lee et al. reported a rate of 87.4% in South Korea [17]. The main reason for this difference may be that tosufloxacin (TFLX), a quinolone agent, has been used to treat children in Japan but not in other countries, including China and South Korea. Ouchi et al. reported that macrolide use for MRMP in children was not as effective as TFLX use [18]. Thus, administration of TFLX may prevent the transmission and spread of MRMP across individuals. Further, the rate of MRMP among MP is low in European countries, such as Germany, possibly because of low antibiotic use, including of macrolides, in European countries. In fact, the proportion of macrolide prescriptions out of the total antibiotic prescriptions is higher in Japan than in European countries [19]. Therefore, macrolide agents are likely to be used more often for *M. pneumoniae* infections in Japan than in European countries. Exposure to antibiotics, especially to low concentrations of macrolides [20], is considered causative of MRMP. Therefore, restricted use of macrolides may be related to decreased rates of MRMP.

Our data indicated a low macrolide-resistance rate for *M. pneumoniae* between 2018 and 2023, which rebounded, however, in 2024. That is, macrolides were less often used because of a decreased case number during that time. Notwithstanding, macrolides comprise the recommended first-line treatment for *M. pneumoniae* infections according to the Japanese guidelines [21]. Therefore, the macrolide-resistance rate for *M. pneumoniae* was almost the same as that during the last endemic period, even in Japan, where TFLX can be used to treat pediatric patients for *M. pneumoniae* infections.

Our study showed that the prevalence of the p1 genotype had changed from the previous endemic period. Kenri et al. reported that the dominant p1 genotype had changed periodically [8] throughout the years, which is similar to our findings. Further, the rates of macrolide resistance markedly varied between the type 1 and type 2 strains. Xiao et al. reported that type 1 strains remained dominant from 2012 to 2018 in the United States [22]. Moreover, the rate of MRMP among type 1 strains was higher than that among type 2 strains. However, the distribution of p1 genotypes reportedly varied across regions, and the rate of MRMP among the total number of strains was low (8.3%). Zhao et al. reported that the dominant genotype was different across regions in China from 2017 to 2018, and the rate of MRMP was higher in type 1 strains than in type 2 strains [23]. Thus, the rate of MRMP tends to be higher in type 1 strains than in type 2 strains, globally.

As previously described, the development of resistance is deeply related to exposure to antibiotics [20], but it is unclear if there are any differences in resistance mechanisms between type 1 and type 2 strains of MP. MRMP was detected for the first time in 2001 in Japan [9], and then increased rapidly throughout Asia [12]. Type 1 strains were dominant during this time period when MRMP was increasing in Japan [11], which was also the same time period of increasing cases of MP infections. There was an increased use of macrolide agents to treat these infections and it is thought that the dominant type 1 strains were exposed to this increased use of macrolides. Therefore, it is possible that if type 2 strains were exposed to the same macrolide use, there may be an increase in MRMP strains that are type 2 in the future. Additionally, p1 types have varied, even within the same time periods, due to different regions in the United States and China [22,23]. Unfortunately, investigation of the p1 genotype with our data is lacking because of the low number of isolates each year. However, it is assumed that the p1 genotypes are not as varied because Japan is a much smaller region than the United States and China.

This study had certain limitations. First, the medical settings where isolates were collected in 2024 were not the same as those in the previous years. However, as already mentioned, a considerable difference may not be observed among the regions given the small size of Japan. Thus, we believe our data can be compared to past data from other areas. Second, detailed backgrounds and the clinical course of the children who provided samples for our research were not analyzed. However, we could compare these data in the future. Second, a limited number of isolates was included in the investigation of the p1 genotypes compared to the total number of isolates. However, we selected them from various areas, and we performed the same analysis each year; thus, we believe that minimal bias was involved. Third, we did not check the MIC of the 16-membered macrolides for *M. pneumoniae*. However, most *M. pneumoniae* isolates with A2063G mutations had low MIC in the past reports [12]. Finally, we examined only P1 genotyping as molecular epidemiology. However, the other molecular methods such as MLVA were employed in a past report [15].

## 5. Conclusions

An epidemic of MP infections had occurred among Japanese children in 2024 for the first time in eight years. Further, the rate of MRMP among these MP isolates was 54.1%, which is similar to the rate observed in the previous epidemic in Japan. However, the dominant *p1* genotype had changed from type 2 in the previous endemic to type 1 in 2024. The rate of MRMP in our study was higher than in European countries, but lower than in other Asian countries, making Japan unique. Thus, it is important to continue monitoring MP infections in Japan.

## Figures and Tables

**Figure 1 microorganisms-13-02243-f001:**
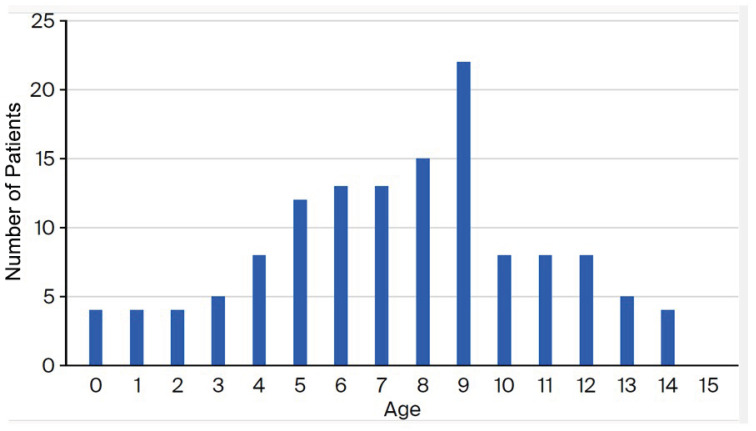
Age distribution of the children from whom MP samples were isolated from in Japan in 2024.

**Figure 2 microorganisms-13-02243-f002:**
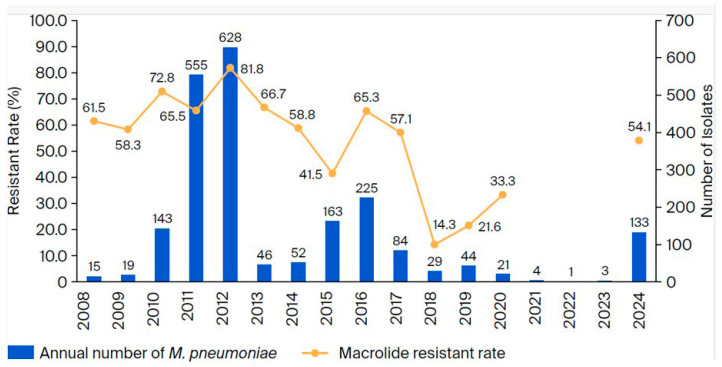
The overall prevalence of MP and the rate of macrolide-resistance among isolates from Japanese children from 2008 to 2024.

**Figure 3 microorganisms-13-02243-f003:**
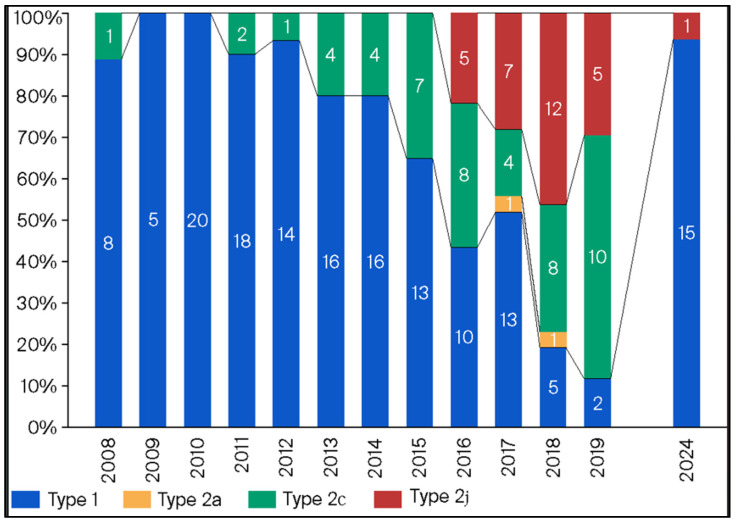
Prevalence of the p1 genotype among MP isolates from 2008 to 2024.

**Figure 4 microorganisms-13-02243-f004:**
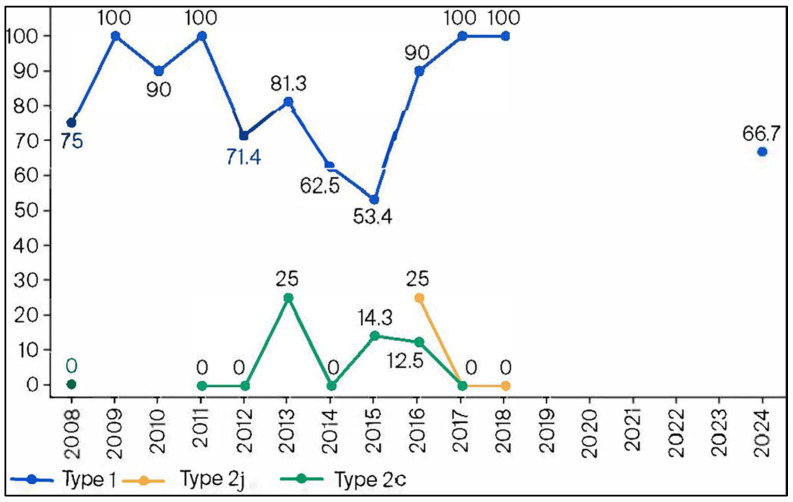
Macrolide-resistance rates according to the p1 genotype from 2008 to 2024.

**Table 1 microorganisms-13-02243-t001:** The distribution of MRMP and mutations associated with macrolide resistance in the 23S rRNA gene isolated from various age groups of Japanese children in 2024.

Age Group (Years)	Number of Isolates	Macrolide-Resistance Mutations in the 23S rRNA Gene			Rate of Macrolide Resistance among Collected *Mycoplasma pneumoniae* Isolates
		No Mutation	A2063G Mutation	C2617A Mutation	
0–4	25	10	15	0	60.0% (15/25)
5–9	75	33	41	1	56.0% (42/75)
10–14	33	18	15	0	45.5% (15/33)
**Total**	**133**	**61**	**71**	**1**	**54.1% (72/133)**

**Table 2 microorganisms-13-02243-t002:** Detection rate of MRMP cases and distribution of point mutations in the 23 rRNA gene in isolates collected from 2008 to 2024.

Year	2008	2009	2010	2011	2012	2013	2014	2015	2016	2017	2018	2019	2020	2024	Total
Detection rate of MRMP (%)	61.5	58.3	72.8	65.5	81.8	66.7	58.8	41.5	65.3	57.1	14.3	21.6	33.3	54.1	69.2
A2063G	8 (100)	7 (100)	99 (100)	303 (94.4)	466 (95.3)	28 (100)	20 (100)	59 (96.7)	115 (100)	32 (100)	3 (100)	12 (100)	7 (100)	71 (98.6)	1229 (96.5)
A2063C	0	0	0	3 (0.9)	0	0	0	0	0	0	0			0	3 (0.2)
A2063T	0	0	0	15 (4.7)	14 (2.9)	0	0	1 (1.6)	0	0	0			0	30 (2.4)
A2064G	0	0	0	0	7 (1.4)	0	0	0	0	0	0			0	7 (0.5)
C2617G	0	0	0	0	2 (0.4)	0	0	0	0	0	0			0	2 (0.2)
C2617T	0	0	0	0	0	0	0	1 (1.6)	0	0	1 (0.1)			1 (1.4)	3 (0.2)

## Data Availability

The data that support the findings of this study are available from the corresponding author, Tomohiro Oishi, upon reasonable request.

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
