# Peer review of "Macrolide-Resistant *Mycoplasma pneumoniae* Among Japanese Children from 2008 to 2024"

_microorganisms, 2025, doi:10.3390/microorganisms13102243_

Round 1
Reviewer 1 Report
Comments and Suggestions for Authors
- in figure 1 the axis name is probably not "age number", but number of patients (absolute number)
- figure 2, in the discussion it is worth discussing the gap between 2018-2023
- line 53 (...85 facilities, 50 clinics, and 35 hospitals...), it is worth indicating how many swabs were collected in total. In 2024, 133 MP strains were isolated from children (77 boys and 56 girls) (line 87), but it is not indicated what % of positive staining, what number of negative staining when it was not possible to isolate a pure culture.
- The manuscript states "We have collected MP samples since 2008" (line 51), and the results of the manuscript cover the time frame 2008-2024, but the title of the manuscript does not reflect this. It is worth noting in the title that the manuscript is about data from 2008 as well.
- a limitation of the study is also the failure to take into account resistance to 16-membered macrolides (e.g. josamycin)
- in the discussion it is worth discussing the reasons for the higher% of resistance than in European countries, perhaps the reason is which antibiotics are most often used for treatment (azithromycin, clarithromycin, etc.).
Author Response
To reviewer 1
Thank you for your comments.
We have revised the manuscript in line with your comments and suggestions and have marked the revisions with tracked changes.
Please find our detailed responses below.
- in figure 1 the axis name is probably not "age number", but number of patients (absolute number)
ÞThank you for your careful review. Accordingly, we have revised the axis label of Figure 1 to “Number of patients.”
- figure 2, in the discussion it is worth discussing the gap between 2018-2023
ÞThank you for this valuable suggestion. We have added relevant text to the discussion following your advice (lines 213-219).
- line 53 (...85 facilities, 50 clinics, and 35 hospitals...), it is worth indicating how many swabs were collected in total. In 2024, 133 MP strains were isolated from children (77 boys and 56 girls) (line 87), but it is not indicated what % of positive staining, what number of negative staining when it was not possible to isolate a pure culture.
ÞThank you for your comment. We have added the number of total samples and total positive samples detected via PCR and culture (lines 98-100).
- The manuscript states "We have collected MP samples since 2008" (line 51), and the results of the manuscript cover the time frame 2008-2024, but the title of the manuscript does not reflect this. It is worth noting in the title that the manuscript is about data from 2008 as well.
ÞThank you for the important advice. We have revised the title from “in 2024” to “from 2008 to 2024.”
- a limitation of the study is also the failure to take into account resistance to 16-membered macrolides (e.g. josamycin)
ÞThank you for noting this. We have included this issue as a limitation (lines 254–256).
- in the discussion it is worth discussing the reasons for the higher% of resistance than in European countries, perhaps the reason is which antibiotics are most often used for treatment (azithromycin, clarithromycin, etc.).
ÞThank you for the useful suggestion. Accordingly, we have added relevant text to the discussion (lines 207-210) as well as reference #20.
Reviewer 2 Report
Comments and Suggestions for Authors
This study describes the national surveillance data of macrolide resistant Mycoplasma pneumoniae in Japan. This manuscript provided valuable epidemiological data regarding an important issue. The manuscript is well-written. I have only a few comments below:
- In introduction section, I recommend briefly reviewing the epidemiology data of M. pneumoniae P1 genotyping in Japan.
- L52: Were samples collected from outpatient clinics or hospitalized patients? Suggest briefly describe the inclusion criteria, exclusion criteria, or the enrollment process.
- Why was only P1 genotyping applied in the study? Why were other methods not used, such as MLVA? Suggest explaining the rationale in method section. And it is also a limitation of this study.
- L87: In line 52, the authors described that the samples were collected from 85 facilities, 50 clinics, and 35 hospitals. So, these 133 isolates were only collected from 8 medical settings? Please ensure the consistency.
- Figure 4: Does Y-axis mean macrolide resistance rate? Suggest clearly label identifiable information in the axis.
- Since tosufloxacin has been used in Japan for years, I am curious about why the macrolide resistance rebound in 2024.
Author Response
To reviewer 2
Thank you for your comments.
We have revised the manuscript in line with your comments and suggestions and have marked the revisions with tracked changes.
Please find my detailed responses below.
In introduction section, I recommend briefly reviewing the epidemiology data of M. pneumoniae P1 genotyping in Japan.
ÞThank you for the valuable suggestion. We have added relevant content (lines 43–44) and reference #11.
L52: Were samples collected from outpatient clinics or hospitalized patients? Suggest briefly describe the inclusion criteria, exclusion criteria, or the enrollment process.
ÞThank you for noting this important issue. We have included a more detailed description of the methodology (lines 59-66).
Why was only P1 genotyping applied in the study? Why were other methods not used, such as MLVA? Suggest explaining the rationale in method section. And it is also a limitation of this study.
ÞThank you for your questions. We have added this as a limitation (lines 256–258).
L87: In line 52, the authors described that the samples were collected from 85 facilities, 50 clinics, and 35 hospitals. So, these 133 isolates were only collected from 8 medical settings? Please ensure the consistency.
ÞWe apologize for the possible misunderstanding. In 2024, the samples were collected from 8 medical settings, 2 clinics and 6 hospitals, in Kyushu, Chugoku-Shikoku, Kinki, and Kanto-Chubu areas in Japan. We have added relevant content (lines 59-60).
Figure 4: Does Y-axis mean macrolide resistance rate? Suggest clearly label identifiable information in the axis.
Þ We apologize for the unclear Y-axis label description. We have corrected it to ”Macrolide resistance rate.”
Since tosufloxacin has been used in Japan for years, I am curious about why the macrolide resistance rebound in 2024.
ÞThank you for your comment. We have added relevant information (lines 213–219) and reference #21.
Round 2
Reviewer 2 Report
Comments and Suggestions for Authors
The authors addressed the issues I mentioned clearly. I just have a minor comment.
In each figure, please ensure all the y-axis and x-axis are properly labelled before publication.
Author Response
To reviewer 2
In each figure, please ensure all the y-axis and x-axis are properly labelled before publication.
Thank you for your suggestion. We asked the author service to edit each figure, and editing completed as you cam see in my article.